# Utility of Real-Time Three-Dimensional Echocardiography for the Assessment of Right Ventricular Morphology and Function in Large Animal Models

**DOI:** 10.3390/jcm11072001

**Published:** 2022-04-02

**Authors:** Yunosuke Yuchi, Ryohei Suzuki, Riho Higuchi, Takahiro Saito, Takahiro Teshima, Hirotaka Matsumoto, Hidekazu Koyama

**Affiliations:** Laboratory of Veterinary Internal Medicine, School of Veterinary Science, Faculty of Veterinary Medicine, Nippon Veterinary and Life Science University, Tokyo 180-8602, Japan; y.0301.yunosuke@gmail.com (Y.Y.); v18061@nvlu.ac.jp (R.H.); justthe2ofussaito@yahoo.co.jp (T.S.); teshima63@nvlu.ac.jp (T.T.); matsumoto@nvlu.ac.jp (H.M.); hkoyama@nvlu.ac.jp (H.K.)

**Keywords:** canine, coupling, ejection fraction, pressure-volume loop, right heart catheterization, right ventricular volume, right ventricular performance, speckle tracking echocardiography, strain, stroke volume

## Abstract

Real-time three-dimensional echocardiography (RT3DE) enables a noninvasive assessment of right ventricular (RV) morphology. However, no study has evaluated the relationship between RV function obtained by RT3DE and RV pressure-volume loops. This hypothesis-driven, experimental study aimed to assess the utility of RT3DE in the evaluation of RV morphology and function. Ten anesthetized beagle dogs sequentially underwent dobutamine infusion, acute infusion of lactated Ringer’s solution, and furosemide administration to alter RV contractility and loading conditions. RV pressure-volume loop-derived hemodynamic measurements and echocardiography, including two-dimensional speckle-tracking echocardiography and RT3DE, were performed in each study protocol. Bland–Altman analysis showed strong agreement in RV volume, ejection fraction, and stroke volume obtained by right heart catheterization and RT3DE. Multiple regression analyses revealed that the peak myocardial velocity of the lateral tricuspid annulus (RV s’) and global RV longitudinal strain rate were significantly associated with end-systolic elastance (adjusted *r*^2^ = 0.66, *p* < 0.001). RV s’, RV free wall longitudinal strain, and RT3DE-derived stroke volume/end-systolic RV volume ratio were associated with RV pressure-volume loops-derived end-systolic/arterial elastance ratio (adjusted *r*^2^ = 0.34, *p* < 0.001). RT3DE could detect the changes in catheterization-derived RV volume with a strong agreement and might be useful in estimating RV-pulmonary arterial coupling.

## 1. Introduction

The right ventricle (RV) has been considered less important to the cardiovascular system for a few decades. However, assessment of its morphology and function has gained attention because various studies have reported the prognostic importance of RV dilatation and dysfunction in patients with various cardiac and pulmonary diseases [1,2]. The gold standards for the evaluation of RV function and morphology are right heart catheterization and cardiac magnetic resonance (CMR) imaging, respectively. Particularly, RV pressure-volume loops obtained by right heart catheterization can provide information on intrinsic RV contractility and RV afterload [1,3]. However, it is impractical to perform these techniques in all cases because of the cost, need for professional equipment, and contraindications. As an alternative, two-dimensional echocardiography (2DE) has been utilized for a simple and quick evaluation of RV morphology and function. However, the crescent-shaped complex geometry of the RV, consisting of inflow, apical, and outflow regions, complicates the entire RV evaluation by 2DE.

Recently, real-time three-dimensional echocardiography (RT3DE) has enabled the noninvasive evaluation of complex RV morphology. This method can measure variables for RV morphology, such as RV volume, ejection fraction (EF), and stroke volume (SV), and was reported to be in agreement with those obtained by CMR [4,5]. Additionally, recent studies have reported the utility of RT3DE-derived ejection fraction (EF) as an RV functional indicator [6,7,8]. However, to the best of our knowledge, no study has compared RV pressure-volume loops and RT3DE in terms of RV functional utility. Additionally, no study has investigated the sensitivity of RT3DE in detecting changes in RV morphology and function with altered RV contractility and loading conditions.

This study aimed to evaluate the agreement of variables obtained by RV pressure-volume loops and RT3DE and investigate the noninvasive independent predictors of RV pressure-volume loop-derived RV functional indicators. We hypothesized that RT3DE could quantify RV morphology with a high agreement with right heart catheterization and detect changes in RV volume that could not be detected by 2DE.

## 2. Materials and Methods

This was a hypothesis-driven, experimental study. All procedures were conducted in accordance with the Guide for Institutional Laboratory Animal Care and Use at Nippon Veterinary and Life Science University and approved by the Ethical Committee for Laboratory Animal Use of Nippon Veterinary and Life Science University, Japan (approval number: 2020S-46).

### 2.1. Animals

Ten laboratory-owned beagle dogs (male/female, 6/4; age, 1.3 ± 0.2 year; body weight, 9.9 ± 0.6 kg) were used in this study. Before enrollment, all dogs were determined to be healthy based on a complete physical examination, complete blood cell count, blood chemistry, thoracic and abdominal radiography, transthoracic and abdominal ultrasonography, and blood pressure measurement.

### 2.2. Study Protocol

All dogs were administered butorphanol tartrate (0.2 mg/kg, IV; Meiji Seika Pharma Co., Ltd., Tokyo, Japan), midazolam hydrochloride (0.2 mg/kg, IV; Maruishi Pharmaceutical. Co., Ltd., Osaka, Japan) and cefazolin sodium hydrate (20 mg/kg, IV; LTL Pharma Co., Ltd., Tokyo, Japan) as preanesthetic medications. Then, they were anesthetized by intravenous administration of propofol (Zoetis Inc., Tokyo, Japan) and maintained with 1.3% isoflurane (Mylan Seiyaku Ltd., Osaka, Japan) mixed with 100% oxygen. The end-tidal partial pressure of carbon dioxide and percutaneous oxygen saturation were monitored and maintained between 35 and 45 mmHg and 98–100%, respectively, using pressure-controlled mechanical ventilation (M&T Corporation, Hiroshima, Japan) at a rate of 10 breaths per minute. Heart rate and blood pressure were monitored throughout the study period. Throughout the study protocol, the dogs received continuous infusions of lactated Ringer’s solution at a rate of 3.0 mL/kg/h.

All anesthetized dogs were placed in the supine position, and the left and right neck regions were clipped, aseptically prepared, and draped. After the injection of lidocaine (Nagase Medicals Co., Ltd., Hyogo, Japan) under the skin around the left and right jugular veins as local anesthesia, a 6-Fr sheath introducer (Terumo Corporation, Tokyo, Japan) was inserted into the left and right jugular veins using the Seldinger retainment technique. The dogs were placed in the left lateral recumbency position, and a 4-Fr pressure-volume catheter (Taisho Biomed Instruments Co., Ltd., Osaka, Japan) and thermodilution catheter (Edwards Lifesciences Corporation, Tokyo, Japan) were positioned in the right ventricle and main pulmonary artery, respectively, with the aid of fluoroscopic guidance (GE Healthcare Japan, Tokyo, Japan). The pressure-volume catheter was manually adjusted to obtain accurate RV pressure-volume loops.

After a 10-min stabilization period, the pressure-volume loop measurement (Σ5 DF Plus; Taisho Biomed Instruments Co., Ltd., Osaka, Japan), cardiac output measurements obtained using the thermodilution technique (LabChart Pro; ADInstruments, Aichi, Japan), and echocardiography (Vivid E95 Ultra Edition with a 4Vc-D (1.7–3.3 MHz) or 6Vc-D (3.1–6.2 MHz) transducer; GE Healthcare, Tokyo, Japan) were performed to obtain baseline data. To increase RV contractility, dogs were continuously administered 5 and 10 µg/kg/min dobutamine for 10 min and underwent the same examination as that at baseline. These two doses of dobutamine were infused sequentially without a washout period. After a 10-min washout period with lactated Ringer’s solution, the same examinations mentioned above were performed again to obtain the baseline data. Acute volume loading was performed by infusing lactated Ringer’s solution at a rate of 150 mL/kg/h for 30 min. Then, the intravenous infusion was stopped, and the dogs were administered furosemide (4.0 mg/kg, IV; Nichi-Iko Pharmaceutical Co., Ltd., Toyama, Japan). The same examinations as those at baseline were performed 15 and 30 min after acute intravenous infusion and furosemide administration. After all the experiments were completed, 1 mL of hypertonic saline was slowly injected through the sheath introducer to estimate parallel conductance.

### 2.3. Hemodynamic Measurement

Right ventricular cardiac output was measured using the thermodilution technique (LabChart Pro; AD Instruments, Aichi, Japan). A lead II electrocardiogram was simultaneously recorded. Dogs were injected with 5 mL iced saline solution from the proximal injectate port three times, and SV was calculated by dividing cardiac output by heart rate obtained using the mean R-R intervals from the same cardiac cycle used for the thermodilution technique. The mean value of three measurements was used to correct the pressure-volume loop.

The pressure-volume loops were analyzed by a single observer (YY) using commercial software (Σ5 DF Plus; Taisho Biomed Instruments Co., Ltd., Osaka, Japan). For each measurement, RV pressure-volume loops and intracardiac ECG data were recorded for 30 s. Additionally, RV pressure-volume loops were corrected using SV obtained from the thermodilution technique and parallel conductance.

The mean values of all hemodynamic variables were calculated using five consecutive pressure-volume loops of sinus rhythm at the end-expiratory phase of mechanical ventilation. From the loops, end-systolic and end-diastolic RV pressures, end-systolic and end-diastolic RV volumes (RVEDV_cath_ and RVESV_cath_, respectively), ejection fraction (EF_cath_), and SV (SV_cath_) were measured. Additionally, the end-systolic elastance (Ees), effective arterial elastance (Ea), and Ees-to-Ea ratio (Ees/Ea) were calculated using the single-beat method, as previously described [9,10].

### 2.4. Two-Dimensional and Doppler Echocardiography

Both 2D and Doppler echocardiographic examinations and measurements were performed by the same observer (YY). A lead II electrocardiogram was recorded simultaneously, and the images were displayed. Data were obtained from three consecutive cardiac cycles in sinus rhythm during the end-expiratory phase of respiration. Images were analyzed using an offline workstation (EchoPAC version 204; GE Healthcare, Tokyo, Japan).

The end-diastolic and end-systolic RV areas (RVEDA and RVESA, respectively) and end-diastolic RV internal dimension (RVIDd) were measured as RV morphology indicators, as previously described [11,12]. The RVIDd was measured as the largest transverse diameter at the middle RV. Additionally, tricuspid annular plane systolic excursion (TAPSE), RV fractional area change (FAC), and peak myocardial velocity of the lateral tricuspid annulus (RV s’) were measured as RV functional indicators, as previously described [11,12,13,14,15]. These indices were obtained using the left apical four-chamber view optimized for the right heart (RV focus view), as previously described [11,12,13,14,15]. TAPSE was measured using the B-mode method [11,16]. The RV s’ were obtained from the tissue Doppler imaging-derived lateral tricuspid annular motion wave.

In this study, two-dimensional speckle-tracking echocardiography (2D-STE) was performed. Right ventricular longitudinal strain and strain rate (RV-SL and RV-SrL, respectively) were obtained from the RV focus view of the same cardiac cycles that were used for echocardiographic measurements of RV morphology and function using left ventricular four-chamber algorithms [10,11,12,17]. The region of interest was defined by manually tracing the RV endocardial border. When necessary, manual adjustments were performed to include the entire RV myocardial wall throughout the cardiac cycle. Only RV free wall analysis (3seg) was performed by tracing the level of the lateral tricuspid annulus to the RV apex. Global RV analysis (6seg) was also performed by tracing from the lateral tricuspid annulus to the septal tricuspid annulus (including the interventricular septum) via the RV apex. The RV-SL was recorded as the absolute value of the negative peak of the strain wave [10,11,12,17,18]. The RV-SrL was recorded as the absolute value of the negative peak of the strain rate wave during systole [10,12].

### 2.5. RT3DE

RT3DE measurements were performed by a single observer (YY) using a commercial application for RV quantification equipped with the echocardiographic system (4D Auto RVQ; GE Healthcare, Tokyo, Japan). Data were obtained from four cardiac cycles of the RV focus view using the multibeat method. From the long-axis and short-axis echocardiographic views displayed automatically, six landmarks, including the free wall and septal tricuspid annulus, RV apex, RV posterior, RV anterior, and RV free wall, were set manually (Figure 1). Then, RV endocardial borders were automatically traced, and manual adjustments were made to obtain accurate echocardiographic variables. In this study, the end-systolic and end-diastolic RV volumes (RVEDV_3D_ and RVESV_3D_, respectively), EF_3D_, and SV_3D_ were measured. Additionally, SV_3D_/RVESV_3D_ was calculated based on previous studies [19]. As these values were obtained using the multi-beat method, the value from a single measurement was considered as the mean value of four cardiac cycles and used in the statistical analysis.

### 2.6. Intra- and Interobserver Measurement Variability

For the RT3DE variables, intra- and inter-observer variabilities were evaluated. Intraobserver measurement variability was assessed by a single observer (YY) using two measurements obtained from the same cardiac cycle and two different days at >7-day intervals. Another blinded observer measured the same variables using the same cardiac cycles for interobserver measurement variability. The coefficient of variation (CV) and intra- or interobserver correlation coefficients (ICC) were calculated to evaluate the intra- and interobserver measurement variability. Low measurement variability was defined as CV < 10.0 and ICC > 0.7. If CV ≥ 10.0 and ICC ≤ 0.7, the data should be judged by a third blinded independent observer. For the other echocardiographic variables than RT3DE, the laboratory intra- and interobserver measurement variabilities have already been reported in our previous studies [11,17,18].

### 2.7. Statistical Analysis

All statistical analyses were performed using the commercially available EZR software version 1.41 (Saitama Medical Center, Jichi Medical University, Saitama, Japan) [20]. All continuous variables are reported as mean ± standard deviation.

For all variables, the Shapiro–Wilk test was performed to evaluate the normality of the data. Each variable was compared between baseline and 5 and 10 µg/kg/min dobutamine infusion and between baseline, 15 and 30 min after acute volume overload, and 15 and 30 min after furosemide administration using repeated-measures analysis of variance with subsequent pairwise comparisons using the Bonferroni-adjusted paired *t*-test for normally distributed data or Friedman rank sum test with subsequent pairwise comparisons using the Bonferroni-adjusted Wilcoxon signed rank sum test for non-normally distributed data.

To evaluate the agreements between RV volume, EF, and SV measured by right heart catheterization and RT3DE, Bland-Altman analysis and Spearman’s rank correlation were performed [21]. Additionally, correlations between RV volume obtained by right heart catheterization and RV morphological variables obtained by 2DE (RV area and RVIDd) were evaluated using Pearson’s or Spearman’s correlation coefficients.

Finally, single and multiple regression analyses were performed to evaluate the association between catheterization-derived Ees and Ees/Ea and echocardiographic variables for RV function. Variables with *p*-values < 0.20 in single regression analyses and adjusted for confounding factors were included in the multiple regression analyses.

In all statistical analyses, *p*-values < 0.05 were considered significant.

## 3. Results

Throughout the experimental procedures, all healthy dogs showed the expected ranges of the end-tidal partial pressure of carbon dioxide (35–45 mmHg), percutaneous oxygen saturation (98–100%), and blood pressure measurements (mean systemic arterial pressure: >60 mmHg). Although carbon dioxide, percutaneous saturation oxygen, and blood pressure level were within the expected ranges noted above, one dog skipped the study protocol of dobutamine infusion because frequent ventricular premature complexes associated with dobutamine infusion complicated the hemodynamic measurements. The mean ± standard deviation of the frame rate in 2D and volume rate in RT3DE were 131 ± 4 and 32 ± 3 frames per second, respectively.

### 3.1. Dobutamine Infusion

Table 1 presents the results of the variables obtained from right heart catheterization, RT3DE, 2DE at baseline, and dobutamine infusions. The end-systolic RV pressure, Ees, Ea, and heart rate significantly increased in a dose-dependent manner. Additionally, EF and SV obtained from right heart catheterization and RT3DE significantly increased with 5 µg/kg/min dobutamine infusion. However, SV alone showed no significant increase with 10 µg/kg/min dobutamine infusion. RVEDV and RVESV measured using the two methods significantly decreased in a dose-dependent manner. SV_3D_/RVESV_3D_ significantly increased with dobutamine infusion. RVESA significantly decreased with 5 and 10 µg/kg/min dobutamine infusions. Additionally, TAPSE, RV FAC, RV s’, RV-SL_3seg_, RV-SL_6seg_, RV-SrL_3seg_, and RV-SrL_6seg_ significantly increased in a dose-dependent manner. The RVEDA and RVIDd showed no significant changes after dobutamine infusion.

### 3.2. Changes in Volume Loading Condition

Table 2 presents the results of the variables obtained from right heart catheterization, RT3DE, and 2DE during the changes in volume-loading conditions. The maximal and minimal RV pressure, RVEDV measured using the two methods, and heart rate significantly increased with acute volume overload and significantly decreased with furosemide administration. Additionally, Ees and Ees/Ea at 15 min after furosemide administration were significantly higher than those at baseline. There were no significant changes in RVESV, EF, SV, Ea, or SV_3D_/RVESV_3D_, although these variables showed similar tendencies. Although RV volume was measured using right heart catheterization and RT3DE, no substantial changes in RVEDA, RVESA, and RVIDd were noted. For the RV functional indices, TAPSE, RV s’, RV-SL_3seg_, RV-SL_6seg_, RV-SrL_3seg_, and RV-SrL_6seg_ significantly increased with acute volume load and significantly decreased with furosemide administration. In contrast, RV FAC showed no significant differences among the changes in volume loading conditions.

### 3.3. Single and Multiple Regression Analysis

The results of the single regression analyses used to estimate Ees and Ees/Ea are summarized in Table 3. RVEDV_3D_, RVESV_3D_, SV_3D_, RVESA, RVIDd, and all 2DE variables, including RV-SL and RV-SrL, showed significant associations with Ees. After adjusting for confounding factors, SV_3D_, RVESA, RV FAC, RV s’, and RV-SrL_6seg_ were included in the multivariate model to estimate Ees, and RV s’ and RV-SrL_6seg_ remained significant in the multivariate model (adjusted *r*^2^ = 0.66, *p* < 0.001):Ees = 0.07 × (RV s’ [cm/s]) + 0.56 × (RV-SrL_6seg_ [%/s])

However, for the Ees/Ea, although there were no significant associations between Ees/Ea and any echocardiographic variables for RV morphology evaluated in this study, EF_3D_, SV_3D_, SV_3D_/RVESV_3D_, RV FAC, RV s’, RV-SL, and RV-SrL_6seg_ showed significant associations with the Ees/Ea. After adjusting for confounding factors, EF_3D_, SV_3D_/RVESV_3D_, RV s’, and RV-SL_3seg_ were included in the multivariate model to estimate Ees/Ea, and RV s’, SV_3D_/RVESV_3D_, and RV-SL_3seg_ remained significant in the multivariate model (adjusted *r*^2^ = 0.34, *p* < 0.001):Ees/Ea = 0.03 × (RV s’ [cm/s]) + 0.30 × (SV_3D_/RVESV_3D_) + (−0.14) × (RV-SL_3seg_ [%])

### 3.4. Agreements between Right Heart Catheterization and RT3DE

Figure 2 and Figure 3 show the results of the scatter plots and Bland–Altman plots of RVEDV, RVESV, EF, and SV. The RVEDV and RVESV obtained by RT3DE showed good agreement between the two measurements (RVEDV, Figure 2a and Figure 3a; RVESV, Figure 2b and Figure 3b). However, these indices had slight but significant fixed errors, showing a higher value in RT3DE (RVEDV, fixed error [95% confidence interval (CI)] = 1.02 [0.72–1.32], *p* < 0.001; RVESV, fixed error [95% CI] = 0.70 [0.28–1.12]). The EF and SV also showed good agreement between the two measurements, and no significant fixed errors were observed in either variable (EF, *p* = 0.268; SV, *p* = 0.140) (EF, Figure 2c and Figure 3c; SV, Figure 2d and Figure 3d). All variables had proportional errors (RVEDV, RVESV, and EF, *p* < 0.001; SV, *p* = 0.014).

Although RVEDA and RVIDd were significantly associated with RVEDV_cath_, the correlations were lower than those with RVEDV_3D_ in this study (RVEDA, *r* = 0.48, *p* < 0.001; RVIDd, *r* = 0.37, *p* < 0.001). Additionally, RVESA showed an intermediate correlation with RVESV (*r* = 0.62, *p* < 0.001), although a strong correlation was observed between RVESV_3D_ and RVESV_cath_.

### 3.5. Intra- and Interobserver Measurement Variability

Table 4 shows the results of the intra- and interobserver measurement variabilities of the RT3DE variables. In this study, RT3DE-derived RV volume, EF_3D_, SV_3D_, and SV_3D_/RVESV_3D_ showed low intraobserver measurement variabilities based on CV < 10.0 and ICC > 0.7. Additionally, low interobserver measurement variability was observed in all these indices.

## 4. Discussion

This is the first study to investigate the clinical utility of RT3DE in RV morphology and functional evaluation in dogs with altered RV contractility and loading conditions. In this study, RT3DE could reflect the RV volume obtained by right heart catheterization with high accuracy and reproducibility, although 2DE-derived RV areas showed no substantial changes associated with dobutamine infusion and acute volume load. These results suggest that RT3DE-derived RV volume can more sensitively detect changes in RV volume. Additionally, EF_3D_, SV_3D_, and SV_3D_/RVESV_3D_ were significantly associated with RV pressure-volume loop-derived Ees/Ea. These variables may be additional tools that reflect RV–pulmonary arterial (PA) coupling (i.e., the balance of RV contractility and RV afterload).

In this study, RVEDV_3D_ and RVESV_3D_ showed strong agreement with RV pressure-volume loop-derived RV volume, which was calibrated using SV obtained from the thermodilution technique and parallel conductance. The RV volume obtained using the two methods significantly increased with acute volume overload and significantly decreased with furosemide administration. Additionally, dobutamine infusion, which would increase RV contractility, decreased not only RVESV but also RVEDV, possibly due to the heart rate-related incomplete relaxation and decrease in venous return. Previous studies have reported good agreement between the RV volume obtained by RT3DE and CMR, the gold standard for cardiac chamber quantification [4,22,23]. Our results also suggest that RT3DE is a feasible, noninvasive tool that could detect the changes in RV morphology associated with altered RV contractility and loading conditions.

Although the RV area and RVIDd obtained by 2DE had similar tendencies to the catheterization-derived RV volume, 2DE variables of the RV inflow region showed only slight changes along with the acute volume load. The main factor leading to these results might be the difficulty in constantly creating the same echocardiographic views. A previous study reported that a slight rotation of the transducer could cause errors in RV chamber measurements in 2DE views [24]. Furthermore, because the RV focus view used in this study could evaluate only the RV inflow region, additional views that could evaluate the other regions would be needed to detect alterations in RV morphology using 2DE. Therefore, although substantial changes in RV volume could be detected by 2DE measurements, including RV area and internal dimension, our results suggest that RT3DE would be more suitable for the quantification of RV morphology.

In the Bland–Altman analyses, almost all baseline data were within the 95% limits of agreement. However, dobutamine infusion and acute volume load caused a dissociation between right heart catheterization and RT3DE in several measurements. These dissociations may be mainly due to increased heart rate. In this study, although the volume rate was increased by whatever means possible, an excessively increased heart rate might complicate detecting the maximal and minimal peaks of RV volume. Therefore, these results should be considered when evaluating patients with heart failure and tachyarrhythmia. Additionally, a single measurement by the multi-beat method for the statistical analyses might have also influenced our results. Repeated measurements might reduce the measurement error. Furthermore, the difficulty tracing the RV endomyocardial border might also lead to the results. Previous studies have reported the difficulty tracing the RV endomyocardial border due to the complex morphology of RV, such as trabeculae carneae and moderator band [13,25,26]. Since RT3DE-derived RV volume is constructed from 2D multi-sectional views, the low reproducibility of RV area might influence our results of RE3DE-derived RV volume. In this study, although RVEDV_3D_ showed excellent agreements with RVEDV_cath_, the measurement agreement of RVESV was not as high as that of RVEDV. Our results indicate that automated RV endomyocardial trace might be difficult, especially in systole, when the narrowed RV cavity and prominent trabeculae carneae are observed.

This is the first study to compare RV function obtained using RT3DE and RV pressure-volume loops, the gold standard for RV functional evaluation [27,28]. Indeed, RV function should be assessed considering RV loading conditions, owing to the load-dependent nature of the RV [29]. As RT3DE-derived functional variables were calculated using the load-dependent RV volume, these indices were considered load-dependent functional indices. Our results suggest that EF_3D_, SV_3D_, and SV_3D_/RVESV_3D_ might enable noninvasive evaluation of RV performance, which reflects RV contractility and RV loading conditions. In this regression analysis, EF_3D_, SV_3D_, and SV_3D_/RVESV_3D_ were significantly associated with Ees/Ea, an indicator of RV-PA coupling (i.e., the balance between RV contractility and RV afterload) [1,3,30]. Previous studies have also reported the prognostic importance of evaluating RV-PA coupling in patients with various cardiac diseases [31,32]. Additionally, other previous studies have reported the close relationship between RV-PA coupling and SV/RVESV obtained by CMR [33,34]. Therefore, SV/RVESV obtained by RT3DE may reflect RV-PA coupling based on Ees/Ea.

Conversely, Ees, the indicator of load-independent RV contractility, was significantly increased with dobutamine infusion and 15 min after furosemide administration, possibly reflecting the increase in RV contractility and force-frequency relationship (Bowditch effect) [35,36]. Although RT3DE variables showed no or a low significant association with Ees, 2D-STE-derived RV-SL and RV-SrL showed the strongest association with Ees. Multiple regression analysis revealed that RV-SrL_6seg_ was an independent factor in estimating Ees. In contrast to RT3DE variables, which reflect load-dependent RV volume, 2D-STE variables were reported to detect precise RV myocardial function with low effects of loading conditions and without angle dependency and translation of the heart [37]. Particularly, a previous study reported that strain rate could be used to evaluate myocardial function regardless of heart rate [38,39]. Therefore, our results suggest that 2D-STE-derived RV-SrL may be more suitable for the detection of intrinsic RV contractility than RT3DE.

This study had several limitations. First, this study used RV pressure-volume loops as the gold standard for RV morphology and function, as an alternative to CMR, which enables the most accurate evaluation of RV morphology. Although volume calibration was performed using the thermodilution technique and parallel conductance, some errors might occur in RV volume measurements compared with CMR. Second, this study used large, healthy animal models. However, since there is no substantial difference in cardiac structure between humans and dogs, similar results could be expected for humans. Third, this study used healthy beagle dogs. Abnormal RV morphology and function might complicate the accurate RV evaluation by RT3DE. Furthermore, there was no control group without giving dobutamine infusion, acute volume overload, and furosemide administration. Anesthesia might influence the results of RV morphology and function. Fourth, this study investigated the utility of RV volume-derived variables only. Additional techniques, such as three-dimensional speckle tracking echocardiography, might be more suitable for the estimation of RV function [40]. Finally, because we could not perform an a priori power calculation, a relatively small sample size might have influenced the statistical power to detect the differences.

## 5. Conclusions

In this study, RT3DE-derived RV volume, EF, and SV detected changes in the RV pressure-volume loops with altered RV contractility and volume loading conditions. Additionally, EF_3D_, SV_3D_, and SV_3D_/RVESV_3D_ were significantly associated with RV pressure-volume loop-derived Ees/Ea. These results support the usefulness of RT3DE in the evaluation of RV morphology and RV-PA coupling. However, higher associations with Ees were observed in 2D-STE-derived RV-SL and RV-SrL compared with RT3DE variables, suggesting that 2D-STE had superior usefulness in the evaluation of intrinsic RV myocardial contractility. Further diagnostic experiments including receiver operating characteristic curve to detect the abnormal RV morphology and function are needed to validate the diagnostic utility of RT3DE in the future.

## Figures and Tables

**Figure 1 jcm-11-02001-f001:**
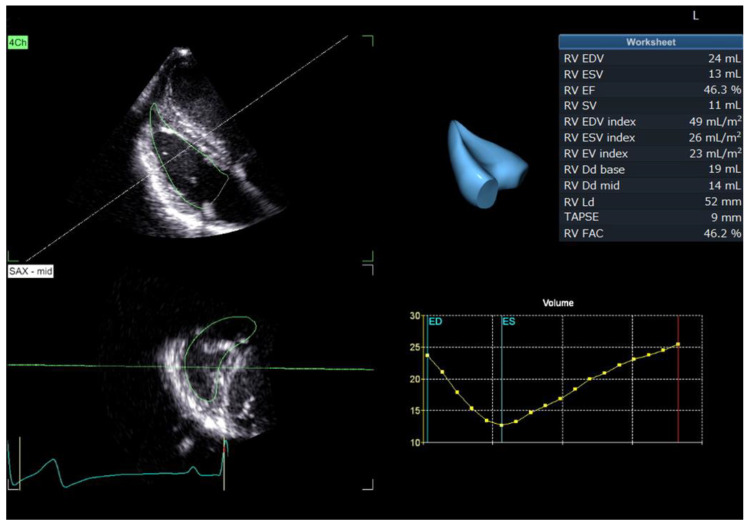
A representative image of the right ventricular volume generated by the 4D Auto RVQ software (GE Healthcare, Tokyo, Japan). 4Ch: left apical four-chamber view optimized for the right heart; ED: end-diastole; ES: end-systole; RV: right ventricular; RVEDV: end-diastolic RV volume; RV EF: RV ejection fraction; RVESV: end-systolic RV volume; RV FAC: RV fractional area change; RV SV: RV stroke volume; SAX–mid: short axis view of the middle right ventricle; TAPSE: tricuspid annular plane systolic excursion. Green circle represents the automatically traced right ventricular endocardial border.

**Figure 2 jcm-11-02001-f002:**
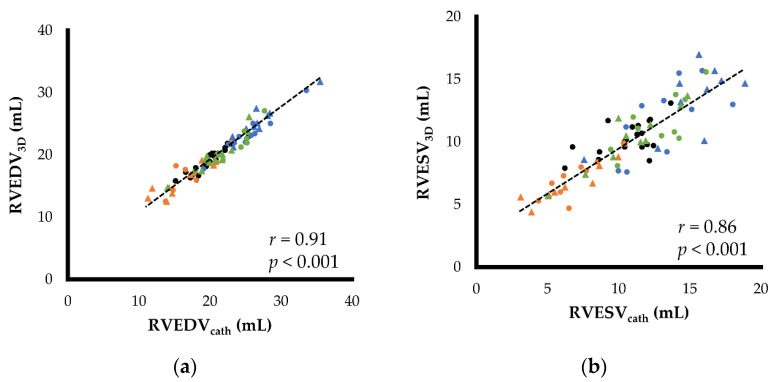
Correlations between the right ventricular morphology and function measured by right heart catheterization and real-time three-dimensional echocardiography: (**a**) end-diastolic right ventricular volume (RVEDV); (**b**) end-systolic right ventricular volume (RVESV); (**c**) stroke volume (SV); (**d**) ejection fraction (EF). Black round dots show the measurements of baseline. Orange round and triangular dots show the measurements of 5 and 10 µg/kg/min dobutamine infusions, respectively. Blue round and triangular dots show the measurements of 15 and 30 min after acute volume load, respectively. Green round and triangular dots show the measurements of 15 and 30 min after furosemide administration, respectively.

**Figure 3 jcm-11-02001-f003:**
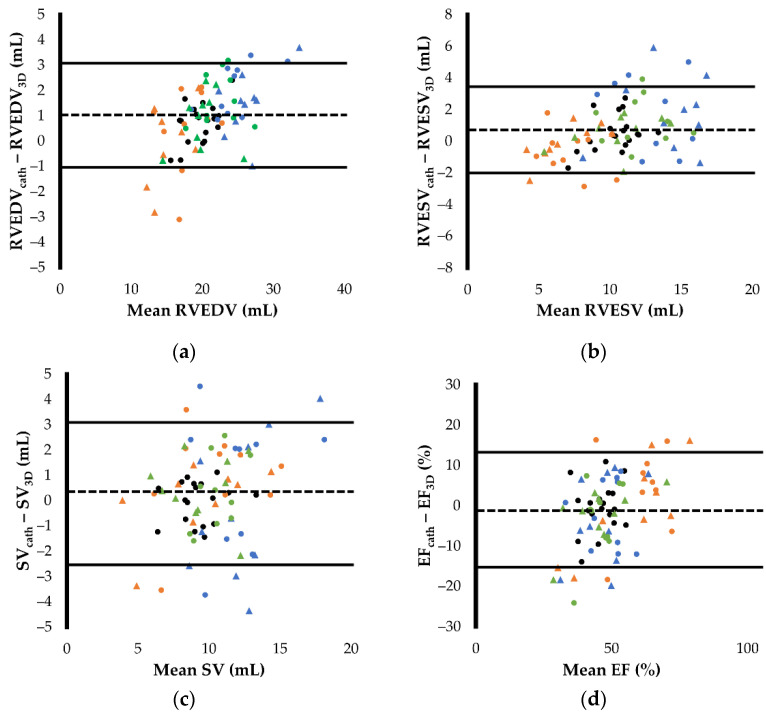
Agreements between the right ventricular morphology and function measured by right heart catheterization and real-time three-dimensional echocardiography demonstrated on Bland–Altman plots: (**a**) end-diastolic right ventricular volume (RVEDV); (**b**) end-systolic right ventricular volume (RVESV); (**c**) stroke volume (SV); (**d**) ejection fraction (EF). The horizontal solid and dotted lines represent the mean difference and the limits of agreement, respectively. Black round dots show the measurements of baseline. Orange round and triangular dots show the measurements of 5 and 10 µg/kg/min dobutamine infusions, respectively. Blue round and triangular dots show the measurements of 15 and 30 min after acute volume load, respectively. Green round and triangular dots show the measurements of 15 and 30 min after furosemide administration, respectively.

**Table 1 jcm-11-02001-t001:** Results of the changes in right ventricular morphology and functional variables obtained by right heart catheterization, real-time three-dimensional echocardiography, and two-dimensional echocardiography with dobutamine infusion.

Variables	Baseline	Dobutamine	
		5 µg/kg/min	10 µg/kg/min
End-systolic RV pressure (mmHg)	12.6 ± 4.2	19.7 ± 3.5 ^a^	23.9 ± 5.5 ^a,b^
End-diastolic RV pressure (mmHg)	3.7 ± 2.3	5.0 ± 1.9	4.1 ± 2.7
RVEDV_cath_ (mL)	20.3 ± 2.3	17.9 ± 3.0 ^a^	16.0 ± 3.4 ^a,b^
RVESV_cath_ (mL)	10.1 ± 3.1	6.9 ± 1.7 ^a^	6.8 ± 2.5 ^a^
EF_cath_ (%)	41.4 ± 14.3	61.6 ± 13.9 ^a^	57.6 ± 20.4 ^a^
SV_cath_ (mL)	8.5 ± 2.9	11.0 ± 3.4 ^a^	9.2 ± 3.6
Ees (mmHg/mL)	1.7 ± 0.2	2.6 ± 0.6 ^a^	4.1 ± 1.2 ^a,b^
Ea (mmHg/mL)	1.6 ± 0.4	2.0 ± 0.7	3.3 ± 1.9 ^a,b^
Ees/Ea	1.2 ± 0.2	1.4 ± 0.3 ^a^	1.4 ± 0.4
RVEDV_3D_ (mL)	19.7 ± 1.9	17.5 ± 2.5 ^a^	16.0 ± 2.6 ^a,b^
RVESV_3D_ (mL)	10.4 ± 1.3	7.1 ± 1.5 ^a^	6.9 ± 1.6 ^a,b^
EF_3D_ (%)	46.1 ± 7.0	58.5 ± 9.4 ^a^	57.7 ± 11.2 ^a^
SV_3D_ (mL)	9.2 ± 1.5	10.3 ± 2.7 ^a^	9.2 ± 2.8
SV_3D_/RVESV_3D_	0.9 ± 0.2	1.5 ± 0.6 ^a^	1.5 ± 0.6 ^a^
Heart rate (bpm)	66.7 ± 11.5	72.3 ± 16.7	83.3 ± 13.6 ^b^
RVEDA (cm^2^)	4.6 ± 0.6	4.6 ± 0.6	4.7 ± 0.9
RVESA (cm^2^)	2.6 ± 0.5	2.0 ± 0.3 ^a^	1.9 ± 0.4 ^a^
RVIDd (mm)	15.2 ± 1.4	15.5 ± 1.4	15.4 ± 1.4
TAPSE (mm)	7.7 ± 1.4	11.6 ± 1.6 ^a^	12.9 ± 1.6 ^a,b^
RV FAC (%)	43.6 ± 3.7	55.7 ± 5.1 ^a^	60.1 ± 2.9 ^a,b^
RV s’ (cm/s)	7.3 ± 1.6	13.0 ± 2.3 ^a^	16.3 ± 3.2 ^a,b^
RV-SL_3seg_ (%)	20.1 ± 1.7	27.9 ± 2.8 ^a^	32.9 ± 3.1 ^a,b^
RV-SL_6seg_ (%)	17.2 ± 1.6	24.3 ± 2.0 ^a^	27.4 ± 2.2 ^a,b^
RV-SrL_3seg_ (%/s)	1.6 ± 0.2	3.5 ± 0.6 ^a^	4.6 ± 1.0 ^a,b^
RV-SrL_6seg_ (%/s)	1.4 ± 0.2	3.1 ± 0.6 ^a^	3.7 ± 0.3 ^a,b^

Continuous data are represented as mean ± standard deviation. 3seg: only RV free wall analysis; 6seg: global RV analysis; Ea: effective arterial elastance; Ees: end-systolic elastance; EF: ejection fraction; RV: right ventricular; RVEDA: end-diastolic RV area; RVEDV: end-diastolic RV volume; RVESA: end-systolic RV area; RVESV: end-systolic RV volume; RV FAC: RV fractional area change; RVIDd: end-diastolic RV internal dimension; RV s’: peak systolic myocardial velocity of the lateral tricuspid annulus; RV-SL: RV longitudinal strain; RV-SrL: RV longitudinal strain rate; SV: stroke volume; TAPSE: tricuspid annular plane systolic excursion. ^a^ The value is significantly different compared with baseline. ^b^ The value is significantly different compared with 5 µg/kg/min.

**Table 2 jcm-11-02001-t002:** Results of the right ventricular morphology and functional variables obtained by right heart catheterization, real-time three-dimensional echocardiography, and two-dimensional echocardiography with acute changes in volume loading conditions.

Variables	Baseline	Acute Volume Overload		Furosemide	
		15 min	30 min	15 min	30 min
End-systolic RV pressure (mmHg)	15.2 ± 2.1	18.6 ± 2.6 ^a^	19.8 ± 3.3 ^a^	15.6 ± 1.4 ^b,c^	13.9 ± 2.1 ^b,c^
End-diastolic RV pressure (mmHg)	3.7 ± 1.1	10.8 ± 1.9 ^a^	10.0 ± 3.6 ^a^	5.7 ± 1.6 ^a,b,c^	3.3 ± 2.0 ^b,c,d^
RVEDV_cath_ (mL)	19.7 ± 2.5	25.4 ± 3.6 ^a^	27.0 ± 3.3 ^a^	23.4 ± 2.7 ^a,c^	20.4 ± 2.9 ^b,c,d^
RVESV_cath_ (mL)	10.2 ± 2.6	13.2 ± 2.5	15.0 ± 3.0	12.8 ± 2.1	10.8 ± 2.8
EF_cath_ (%)	45.4 ± 12.4	47.7 ± 7.8	44.6 ± 11.7	44.4 ± 8.4	44.6 ± 13.5
SV_cath_ (mL)	9.6 ± 2.0	12.2 ± 2.9	12.0 ± 3.4	10.7 ± 1.7	9.7 ± 2.3
Ees (mmHg/mL)	1.6 ± 0.2	1.8 ± 0.3	1.9 ± 0.4	2.0 ± 0.5 ^a^	1.7 ± 0.4
Ea (mmHg/mL)	1.8 ± 0.5	1.5 ± 0.3	1.7 ± 0.7	1.6 ± 0.5	1.6 ± 0.6
Ees/Ea	1.0 ± 0.3	1.3 ± 0.3	1.2 ± 0.3	1.3 ± 0.2 ^a^	1.2 ± 0.5
RVEDV_3D_ (mL)	18.9 ± 2.0	23.5 ± 3.3 ^a^	25.5 ± 2.7 ^a^	21.8 ± 2.6 ^a,c^	19.6 ± 2.7 ^b,c,d^
RVESV_3D_ (mL)	10.0 ± 1.3	11.9 ± 2.8	13.1 ± 2.5	11.5 ± 2.1	10.2 ± 2.3 ^c^
EF_3D_ (%)	46.3 ± 6.6	49.6 ± 8.5	48.2 ± 8.3	47.5 ± 5.0	46.8 ± 9.4
SV_3D_ (mL)	8.9 ± 1.9	11.6 ± 2.2	12.5 ± 2.4	10.3 ± 1.2	9.4 ± 2.1
SV_3D_/ RVESV_3D_	0.8 ± 0.3	1.1 ± 0.4	1.0 ± 0.3	1.0 ± 0.2	1.0 ± 0.4
Heart rate (bpm)	79.3 ± 8.1	96.0 ± 15.7	125.9 ± 29.2 ^a^	110.5 ± 32.5 ^a^	93.7 ± 21.4
RVEDA (cm^2^)	4.7 ± 0.5	5.1 ± 0.5	5.1 ± 0.7 ^a^	5.0 ± 0.4	4.7 ± 0.5
RVESA (cm^2^)	2.7 ± 0.4	2.9 ± 0.5	2.9 ± 0.4	2.7 ± 0.2	2.8 ± 0.3
RVIDd (mm)	15.6 ± 1.1	16.2 ± 1.0	16.5 ± 1.3	16.2 ± 1.3	15.6 ± 0.9
TAPSE (mm)	8.3 ± 0.9	11.7 ± 1.4 ^a^	12.8 ± 1.4 ^a^	10.7 ± 1.1 ^a,c^	9.3 ± 1.6 ^b,c,d^
RV FAC (%)	43.3 ± 4.1	42.8 ± 5.7	43.6 ± 4.8	46.1 ± 3.5	41.2 ± 3.8
RV s’ (cm/s)	7.7 ± 1.4	9.7 ± 1.2 ^a^	10.5 ± 1.2 ^a^	10.0 ± 1.5 ^a^	9.0 ± 1.7
RV-SL_3seg_ (%)	21.4 ± 2.5	24.6 ± 1.8	26.4 ± 2.7 ^a^	24.2 ± 3.2	22.2 ± 3.1 ^c^
RV-SL_6seg_ (%)	17.3 ± 2.1	20.5 ± 1.4 ^a^	21.7 ± 1.7 ^a^	19.7 ± 2.3 ^a^	17.7 ± 2.4 ^b,c,d^
RV-SrL_3seg_ (%/s)	1.9 ± 0.4	1.9 ± 0.3	2.4 ± 0.5 ^b^	2.2 ± 0.7	1.8 ± 0.4 ^c^
RV-SrL_6seg_ (%/s)	1.5 ± 0.3	1.6 ± 0.2	1.9 ± 0.4 ^a^	1.8 ± 0.6	1.5 ± 0.3 ^c^

Continuous data are represented as mean ± standard deviation. 3seg: only RV free wall analysis; 6seg: global RV analysis; Ea: effective arterial elastance; Ees: end-systolic elastance; EF: ejection fraction; RV: right ventricular; RVEDA: end-diastolic RV area; RVEDV: end-diastolic RV volume; RVESA: end-systolic RV area; RVESV: end-systolic RV volume; RV FAC: RV fractional area change; RVIDd: end-diastolic RV internal dimension; RV s’: peak systolic myocardial velocity of the lateral tricuspid annulus; RV-SL: RV longitudinal strain; RV-SrL: RV longitudinal strain rate; SV: stroke volume; TAPSE: tricuspid annular plane systolic excursion. ^a^ The value is significantly different compared with baseline. ^b^ The value is significantly different compared with 15-min acute volume overload. ^c^ The value is significantly different compared with 30-min acute volume overload. ^d^ The value is significantly different compared with 15 min after furosemide administration.

**Table 3 jcm-11-02001-t003:** Results of single regression analyses to estimate right heart catheterization-derived right ventricular functional variables.

Variables	Ees		Ees/Ea	
	Regression Coefficient (95% CI)	*p*	Regression Coefficient (95% CI)	*p*
RVEDV_3D_ (mL)	−0.1 (−0.15–−0.06)	<0.001	0.02 (0.00–0.04)	0.079
RVESV_3D_ (mL)	−0.11 (−0.17–−0.04)	0.002	−0.01 (−0.04–0.01)	0.360
EF_3D_ (%)	0.01 (−0.02–0.03)	0.641	0.01 (0.01–0.02)	<0.001
SV_3D_ (mL)	−0.10 (−0.18–−0.02)	0.011	0.06 (0.04–0.09)	<0.001
SV_3D_/RVESV_3D_	0.20 (−0.24–0.65)	0.371	0.33 (0.18–0.48)	<0.001
RVEDA (cm^2^)	−0.19 (−0.51–0.13)	0.246	0.07 (−0.05–0.20)	0.219
RVESA (cm^2^)	−0.93 (−1.27–−0.59)	<0.001	−0.09 (−0.24–0.06)	0.239
RVIDd (mm)	−0.17 (−0.32–−0.01)	0.033	0.02 (−0.04–0.08)	0.416
TAPSE (mm)	0.17 (0.09–0.25)	<0.001	0.03 (−0.01–0.06)	0.111
RV FAC (%)	0.08 (0.06–0.10)	<0.001	0.01 (0.00–0.02)	0.006
RV s’ (cm/s)	0.19 (0.15–0.24)	<0.001	0.04 (0.02–0.06)	0.001
RV-SL_3seg_ (%)	0.13 (0.09–0.16)	<0.001	0.02 (0.00–0.03)	0.038
RV-SL_6seg_ (%)	0.16 (0.12–0.20)	<0.001	0.02 (0.00–0.04)	0.041
RV-SrL_3seg_ (%/s)	0.61 (0.49–0.74)	<0.001	0.06 (−0.01–0.13)	0.099
RV-SrL_6seg_ (%/s)	0.81 (0.67–0.96)	<0.001	0.09 (0.01–0.18)	0.031

3seg: only RV free wall analysis; 6seg: global RV analysis; CI: confidence interval; Ees: end-systolic elastance; Ees/Ea: ratio of Ees and effective arterial elastance; EF: ejection fraction; RV: right ventricular; RVEDA: end-diastolic RV area; RVEDV: end-diastolic RV volume; RVESA: end-systolic RV area; RVESV: end-systolic RV volume; RV FAC: RV fractional area change; RVIDd: end-diastolic RV internal dimension; RV s’: peak systolic myocardial velocity of the lateral tricuspid annulus; RV-SL: RV longitudinal strain; RV-SrL: RV longitudinal strain rate; SV: stroke volume; TAPSE: tricuspid annular plane systolic excursion.

**Table 4 jcm-11-02001-t004:** Results of the intra- and inter-observer measurement variabilities in real-time three-dimensional echocardiography variables.

Variables	Intra-Observer		Inter-Observer	
	CV (%)	ICC	CV (%)	ICC
RVEDV_3D_ (mL)	4.2	0.93 *	8.1	0.89 *
RVESV_3D_ (mL)	3.9	0.96 *	6.9	0.90 *
EF_3D_ (%)	5.2	0.89 *	3.7	0.92 *
SV_3D_ (mL)	8.1	0.90 *	9.9	0.81 *
SV_3D_/RVESV_3D_	9.6	0.82 *	6.6	0.90 *

CV: coefficient of variation; EF: ejection fraction; ICC: intra- or inter-observer correlation coefficient; RV: right ventricular; RVEDV: end-diastolic RV volume; RVESV: end-systolic RV volume; SV: stroke volume. * Within a row, ICC values were considered significant (*p* < 0.05).

## Data Availability

The datasets used or analyzed during the current study are available from the corresponding author upon reasonable request.

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
