# Peer review of "Utility of Real-Time Three-Dimensional Echocardiography for the Assessment of Right Ventricular Morphology and Function in Large Animal Models"

_jcm, 2022, doi:10.3390/jcm11072001_

Round 1

Reviewer 1 Report

The study Utility of real-time three-dimensional echocardiography for the assessment of right ventricular morphology and function in large animal models has evaluated the relationship between RV function obtained by RT3DE and RV pressure-volume loops and mainly concluded that RT3DE could detect the changes in catheterization-derived RV volume with strong agreement. This original work may provide an interesting novel method for noninvasive assessment of right ventricular function. However, the manuscipt is still lacking in the designing of protocol, the presentation of the experimental results and the justification of the conclusions, and further revisions are required.

Specific comments:

Materials and Methods:

  1. (Line17-19) RV pressure-volume loop-derived hemodynamic measurements and echocardiography including teo-dimensional (should be: two-dimensional) speckle-tracking echocardiography...
  2. (Line 100, 132+) There wereno transducer parameters marked in the echocardiographic part, so it was unclear whether it is suitable for applying in dog 

Example:

Two‐dimensional echocardiography (2DE) was carried out in both groups using an iE33 echocardiographic system (Philips Medical Systems, Andover, MA) with an S5-1 transducer (1-5 MHz) Real time three-dimensional echocardiography (RT3DE) was performed using iE33 echocardiographic system equipped with an X3-1 transducer (1-3 MHz) in both gruops.

  1. (Line 102) 5 and 10 µg/kg/min dobutamine for 10 min....Were these two doses of dobutamine used sequentially? Was there a washout period in between? It was not clearly described in the manuscript.
  2. (Line 105-106) Acute volume loading was performed by infusing lactated Ringer's solution at a rate of 150 mL/kg/h for 30 min.... It wasunclear whether there was a washout period between the injection of lactated Ringer's solution and the injection of dobutamine previously. If not, dobutamine might affect morphology and functional variables in Table 2.
  3. (Line 108-109) The same examinations as those at baseline were performed 15 and 30 min after acute intravenous infusion and furosemide administration. What is the basis for determining the two time points of 15 and 30 minutes? The action time of furosemide in the human body can be maintained for 2 hours. Should multi-group time-effect relationships be considered in the study protocoland extended the observation period to 2 hours?
  4. (Line 133) Conventional and Doppler echocardiographic examinations and measurements.... Here, “conventioal”could be change into “2D”.
  5. (Line 140) end-diastolic RV internal dimension (RVIDd)....RVIDd could be measured in several sub-segments (basal, middle...) in RV, which typical segment was measured in the study? Please describe them in detail.

Example:

Transverse diameter of right ventricular basal segment (RVD1) and middle segment (RVD2) were collected for analysis.

  1. (Line 163) RV quantification (4D Auto RVQ; GE Healthcare) ....The software information wasnot detailed enough, please add the version number.

Example: TomTec 4D LV analysis (4.6.0.411, TomTec Imaging Systems GMBH, Germany)...

  1. (Line 175-176) ...generated by the right ventricular quantification software... please change “quantification software”into “4D Auto RVQ software” to be more specific.
  2. (Line 175-176) The coefficient of variation (CV) and intra- or interobserver correlation coefficients (ICC) were calculated to evaluate the intra- and interobserver measurement variability. Low measurement variability was defined as CV < 10.0 and ICC > 0.7.

Could add: If CV ≥ 10.0 and ICC ≤ 0.7, the data should be judged by a third blinded independent observer.

  1. (Line 200-201) ....correlations between RV volume obtained by right heart catheterization and RV morphological variables obtained by 2DE (RV area and RVIDd).....RT3DE has RVEDV3Dand RVESV3D could be evaluated RT3DE, why not use thse for analysis ?

Results

  1. (Line 209) In this study, one dog skipped the study protocol of dobutamine infusion... Please add some detailed information, likecarbon dioxide, percutaneous saturation oxygen, and blood pressure level at the time of the dog skipped the study.
  2. (Line 211-213) The end-tidal partial pressure of carbon dioxide, percutaneous oxygen saturation, and blood pressure measurements were within the expected ranges in healthy dogs throughout the experimental procedures. What was the expected range ? Values for carbon dioxide, percutaneous oxygen saturation, and blood pressure should be described.
  3. (Line 249-250)Although RV volume was measured using right heart catheterization and RT3DE, no substantial changes in RVEDA, RVESA, and RVIDd were noted.... RVIDd could be detailed evaluated in segments (see Materials and Methods:7)
  4. (Line 266) a.The value is significantly different compared with baseline. ...The legend has two a, please change it into a proper word.
  5. (Line 271-294, part 3.3) Now that the regression coefficient waslisted, the regression equation couldbe listed simultaneously.
  6. (Line 304-305) All variables had proportional errors (maximal and RVESV; EF, p < 0.001; SV, p = 0.014). What didit mean?Might be change into “maximal are RVESV, p < 0.001; EF, p < 0.001;...”
  7. (Line 314-315,Fig 2) (c) ejection fraction (EF); (d) stroke volume (SV) should be “(c) stroke volume (SV); (d) ejection fraction (EF)”
  8. (Line 315-316, Fig 2) Green, orange, and blue dots show the measurement values ​​of baseline, dobutamine infusion, and acute volume load, respectively....dobutamine infusion wasin two doses (5 µg/kg/min, 10 µg/kg/min), acute volume overload had15 min and 30 min data, and after Furosemide infusion there were 15 min and 30 min data, all of them could be represented respectively by dots with different colors/shapes.
  9. (Line 321, Fig 3) (c) ejection fraction (EF); (d) stroke volume (SV)..should be “(c) stroke volume (SV); (d) ejection fraction (EF)”.
  10. (Line 322-234, Fig 3) Blue, orange, and green dots show the measurement values of baseline, dobutamine infusion, and acute volume load, respectively. (Similar topiont8) dobutamine infusion was in two doses (5 µg/kg/min, 10 µg/kg/min), acute volume overload had 15 min and 30 min data, and after Furosemide infusion there were 15 min and 30 min data, all of them could be represented respectively by dots with different colors/shapes.
  11. Only using multiple regression analyses to prove that "RT3DE could detect the changes in catheterization-derived RV volume with strong agreement" is not sufficient, further diagnostic experiments should be designed, ROC curves should be made, and AUC should be reported. This paper lackedsuch experimental design, so the credibility of the conclusions is questionable.

Discussion

  1. (Line 365-366) .......evaluate only the RV inflow region, additional views that could evaluate the other regions would be needed....The echocardiography foucus more in RV inflow region, the situation of RV outflow regionremained unclear, saying “variables showed only slight changes along the acute volume load (Line 361-362)” was no adjusted. Consider changing to "2DE variables of RV inflow region showed only slight changes..."
  2. (Line 372) However, dobutamine infusion and acute volume load caused a dissociation between right heart catheterization and RT3DE in several measurements.....Since "tracing the RV endomyocardial border might be difficult" (Line 376-377), whether applyingRepeated measurements to reduce measurement error?
  3. The sample size n=10was too In order to exclude the influence of anesthesia itself on RV parameters, a control group without giving dobutamine infusion, acute volume overload, and furosemide infusion.

Author Response

Dear Reviewer 1

We wish to express our strong appreciation to the Reviewer for their insightful comments on our paper. We feel the comments have helped us significantly improve the paper. We hope that the revised paper meets your approval and will be more suitable for publication in the Journal of Clinical Medicine as Original Research. Please see the attachment file for details.

Reviewer 2 Report

The authors in their paper report an interesting analysis on the role of real time 3DSTE in a dog sample. They found a strong correlation between invasive hemodynamic data and non-invasively measured echocardiographic parameters. The paper is of interest, with a potential clinical impact if translated on humans.

I have the following concerns.

Major points

  • The sample size is quite small. This limitation should be acknowledged
  • In humans, 3DSTE showed to be very helpful in patients with arterial hypertension and, especially, in atrial arrhythmias even during sinus rhythm (doi: 10.1038/s41598-019-43855-7). This point should be discussed
  • I see that the strongest agreement between non-invasive and invasive assessment is on RVEDV, while RVESV and RFEV show a less strong agreement. Some discussion on this result should be provided.

Minor points

  • Please report the sequence of ringer and furosemide infusion in the abstract
  • In the abstract, please report some relevant numbers (e.g. p values, r values, descriptive analyses)
  • Laboratory intra and interoperator reproducibility should be performed or previous experiences showing this reproducibility should be cited.
  • Cardiac MR findings are known to be in correlation with hemodynamic parameters. The lack of cardiac MR is a relevant acknowledged as a limitation.

Author Response

Dear Reviewer 2

We wish to express our strong appreciation to the Reviewer for their insightful comments on our paper. We feel the comments have helped us significantly improve the paper. We hope that the revised paper meets your approval and will be more suitable for publication in the frontiers in Veterinary Science as Original Research. Please see the attachment file for details.
